# OpenReview forum: "Peacemaker or Troublemaker: How Sycophancy Shapes Multi-Agent Debate"
_ICLR.cc/2026/Conference — ICLR 2026 Conference Withdrawn Submission_

### Official Review · Reviewer_A1tX · 2025-10-31

**Soundness:** 2
**Presentation:** 3
**Contribution:** 2
**Rating:** 4
**Confidence:** 4

**Summary:**

The paper studies sycophancy in multi-agent debates by
(i) giving an operational definition tailored to MADS,
(ii) introducing three metrics: Disagreement Collapse Rate (DCR), Negative Agreement Rate (NAR), and Sycophancy Score (SS)
(iii) controlling debater/judge personas along a peacemaker↔troublemaker spectrum.
Experiments span decentralized (SoM) and centralized (judge) setups implemented in AutoGen, with Qwen3-32B and LLaMA-3.3-70B on CommonsenseQA and MMLU-Pro. The core finding is that sycophancy is tightly linked to disagreement collapse; multi-agent performance often fails to beat single-agent baselines, and persona mixing (one more independent, one more cooperative) tends to outperform uniformly agreeable teams, while judge sycophancy has limited effect in centralized settings.

**Strengths:**

1. Introduce new metrics. The introduction of disagreement collapse rate (DCR), negative agreement rate (NAR), and sycophancy score (SS) sharpens quantitative analysis of system behavior, and these metrics are clearly motivated and mathematically specified.
2. The framework for inducing and tuning agent sycophancy via explicit persona prompts is methodologically sound and allows fine-grained experimental exploration of agent behaviors.
3. Conduct relatively comprehensive experiments to illustrate their conclusion.

**Weaknesses:**

- Manipulation check for persona levels. The paper assumes λ=1→8 induces increasing sycophancy, but doesn’t show a simple trend test (e.g., SS vs. λ monotonicity) or robustness to prompt paraphrases/order shuffles.

- Evaluator-driven SS. SS relies on an LLM grader (GPT-5-mini). Please include a small human audit and inter-rater agreement, or at least cross-model grader agreement, to bound evaluator bias.

- Limited coverage of tasks/models/scale. Results are on two QA datasets with two backbones; Qwen thinking is disabled; agent counts are 2–3 due to cost. Broader tasks (tool-use, long-horizon planning), “thinking” variants, and larger agent pools would strengthen generality

- Cost vs. benefit. The paper notes gains are modest relative to overhead; please add token/latency budgets and accuracy-per-token curves vs. strong single-agent baselines (self-consistency, diverse CoT).

**Questions:**

- Can you report SS–λ monotonicity and a robustness test under paraphrased persona prompts?

- Where is the boundary between sycophancy and conformity? Without a clear separation, the work risks overlapping with the LLM conformity literature.

- Are these short-form tasks already dated for the agent era? In agentic settings we care more about long-horizon, tool-using workflows. The current debate protocol may not scale to complex tasks for at least two reasons:
a) Complex problems demand sustained reasoning/retrieval, and multi-round debate quickly exhausts the context/token budget.
b) The debate objective is under-specified: what exactly are agents optimizing—raw accuracy, novelty, evidence diversity, calibrated confidence, or progress from PD→PA? Please make this explicit.

---

> ### Author Response · Authors · 2025-12-03
> **Reply to Reviewer A1tX**
>
> Dear Reviewer A1tX,
>
> Thanks for your insightful reviews and suggestions. We appreciate your recognition of soundness for our sycophancy tuning method. We believe many of your concerns are addressable in the revised version.
>
> 1. **Manipulation check for persona levels**: Our experiments indeed assume that larger λ induces stronger sycophancy, and we agree that validating this relationship empirically would strengthen the work. Our persona prompt design follows prior work showing that scaling λ from 1→8 increases sycophantic behaviors [1]. In the revision, we will report the monotonic trend between λ and SS, as well as provide robustness checks under paraphrased and order-shuffled persona prompts.
> 2. **Evaluator-driven SS**: We agree that relying on GPT-5-mini as the sole evaluator may introduce grader-specific bias. We will add cross-model evaluation using at least one additional LLM grader, reported in the appendix, to bound evaluator variance. These agreement checks will increase the reliability of SS as an operational measurement.
> 3. **Limited coverage of datasets, models, and scale**: We acknowledge that broader empirical coverage could further strengthen the study. Our objective, however, is not to provide general system design principles across all possible settings, but to introduce an operational framework for analyzing and designing debating systems through the lens of sycophancy, which is a factor that our results show to be strongly correlated with failures in handling disagreement within debate. Prior work has demonstrated that debate behaviors can vary substantially across model versions, datasets, and implementations [1]. Accordingly, our contribution focuses on isolating sycophancy as a key tunable parameter rather than attempting an exhaustive survey of every configuration.
> Our empirical choices were made to ensure representativeness while maintaining a clear analytic focus. We evaluate (1) two task types—reasoning and factuality—where debate is known to outperform single-agent generation, (2) two widely used open-source models from distinct model families (Qwen and Llama), and (3) two canonical debate paradigms that form the basis of many later variants. This design enables us to study sycophancy within the foundational communication structures of debate, without confounds introduced by highly specialized or rapidly evolving systems. Across all examined settings, we consistently find that sycophancy correlates with disagreement collapse and associated performance degradation, and that controlling sycophancy yields measurable improvements. We will release our evaluation and sycophancy-control code to support application of the framework to additional models, datasets, and debate frameworks.
> 4. **Cost vs. benefit**: We acknowledge the reviewer’s concern regarding modest gains relative to multi-agent overhead. Our goal is not to claim state-of-the-art accuracy, but to show that sycophancy is a key factor driving disagreement collapse and performance variability in debate systems. In the revision, we will consider adding explicit token and latency budgets in the appendix, and providing accuracy-per-token curves comparing controlled MADS to strong single-agent baselines such as self-consistency and diverse CoT. However, this should have no detrimental impact on the contributions of our work, proposing an operational framework for analyzing and designing debating systems through the lens of sycophancy
> 5. **Boundary between sycophancy and conformity**: In our framework, sycophancy refers to directional preference adaptation toward a specific peer agent without independent reasoning, whereas conformity reflects general alignment with majority or normative behaviors independent of a specific interlocutor. Sycophancy is thus a relational, agent-to-agent property, while conformity is a population-level or norm-level phenomenon. Our measurement specifically targets the former by comparing agent responses to detect direct echoing or dependence.
> 6. **Applying to long-horizon, tool-using tasks**: We agree these tasks are increasingly important. However, our goal in this paper is to establish a principled, model-agnostic framework for quantifying and controlling sycophancy, which is an issue that becomes even more critical as agent workflows lengthen. Moreover, we will clarify that our framework does not assume a fixed debate objective. It is compatible with goals such as accuracy, evidence diversity, calibrated confidence, or progress from PD→PA. Extending the framework to long-horizon and tool-using settings is a natural and important next step, and we view this work as laying the foundation for those extensions.
>
> Thank you again for your constructive feedback!
>
> [1] Chen, Runjin, et al. "Persona vectors: Monitoring and controlling character traits in language models." arXiv preprint arXiv:2507.21509 (2025).

---

### Official Review · Reviewer_c3AJ · 2025-10-31

**Soundness:** 3
**Presentation:** 3
**Contribution:** 3
**Rating:** 6
**Confidence:** 4

**Summary:**

This paper studies how LLM sycophancy—excessive agreement—disrupts multi-agent debate (MAD) by collapsing productive disagreements and degrading accuracy below single-agent baselines. It formalizes sycophancy for both debaters and judges, and introduces quantitative metrics: Disagreement Collapse Rate (DCR) at the system level, Negative Agreement Rate (NAR) at the agent level, and a Sycophancy Score (SS) that distinguishes independent reasoning from echoing peers. The authors evaluate decentralized Society-of-Minds (peer-to-peer) and centralized MAD (two debaters + judge) frameworks, finding that sycophancy induces persistent collapse overall and that centralized judging reduces but does not eliminate it. Beyond measurement, the paper proposes prompt-based persona control over a discrete “troublemaker↔peacemaker” spectrum and performs grid searches, showing that mixing independent and conciliatory personas improves steerability while limiting collapse, especially in heterogeneous debates.

**Strengths:**

* This paper highlights sycophancy in current MAD systems as the key failure mode.
* The authors propose a novel method to measure the sycophancy in MAD systems, Sycophancy Score, based on the behavior of agents in MAD.
* The emergence of sycophancy is also discussed, supported by comprehensive evaluation and anaylsis.

**Weaknesses:**

* While with in-depth analysis and design recommendations, there is not a unified solution to the problem, which weakens the technical contirbution of this paper.
* In the section "Controlled Sycophancy", the evaluation relies heavily on persona vectors, which goes beyond the normal scenarios where foundation models are direcly used with approporiate prompting in MAD systems. It may be more interesting and helpful to measure the sycophancy in some pracrical scenarios.

**Questions:**

N/A

---

> ### Author Response · Authors · 2025-12-03
> **Reply to Reviewer c3AJ**
>
> Dear Reviewer c3AJ,
>
> Thanks for your insightful and positive feedback on our work, we’ll revise our paper based on your comments carefully.
>
> 1. **The lack of a unified solution**: Our goal is not to propose a single universal algorithm for eliminating sycophancy across all multi-agent debate systems. Instead, the contribution of this work is to provide an operational framework that makes sycophancy measurable, controllable, and integrable into debate-system design. Given the substantial variation in debate behavior across models, datasets, and architectures reported in prior work, prescribing a one-size-fits-all solution would be neither realistic nor robust. The value of our contribution lies in identifying the underlying mechanisms, such as disagreement collapse, and introducing sycophancy as a tunable parameter for analyzing and designing multi-agent debate, rather than offering a fixed design principle intended to generalize across all settings.
> 2. **The use of persona vectors in controlled sycophancy**: We use persona vectors as a systematic and interpretable way to expose and modulate sycophancy levels under controlled conditions, enabling us to isolate its impact on debate dynamics. Importantly, our implementation follows the prompting-based multi-agent debate setup used in prior work (in Appendix C), and all sycophancy control is performed solely through the system prompt without altering task instructions.
>
> We thank you once again for your valuable feedback!

---

### Official Review · Reviewer_cbtG · 2025-11-01

**Soundness:** 2
**Presentation:** 3
**Contribution:** 2
**Rating:** 4
**Confidence:** 4

**Summary:**

This paper investigates how sycophancy impacts multi-agent debating systems (MADS). It introduces a framework to define, measure, and control sycophancy in MADS by adjusting agent personas from troublemaker to peacemaker. Experiments show sycophancy causes disagreement collapse, leading to poor performance, sometimes worse than single agents. The study finds that balancing personas is key, and centralized systems are more robust to judge sycophancy.

**Strengths:**

The paper systematically addresses the underexplored issue of inter-agent sycophancy in MADS. The findings on persona dynamics are interesting, particularly the conclusion that mixing peacemaker and troublemaker roles yields better results than minimizing sycophancy uniformly.

**Weaknesses:**

1. Limited Dataset Diversity: The narrow dataset scope (MMLU Pro, CommonsenseQA) may not capture the full picture of how sycophancy affects MADS across diverse task types (e.g., math, code), limiting the generalizability of the findings.

2. Missing Comparison to Self-Consistency (SC): The paper compares MADS only to a single-agent baseline, which is insufficient given the high computational cost of MADS (potentially 15 LLM calls per query vs. 1 call). Failing to compare against SC with a comparable computational budget (e.g., SC sampling 15 times) leaves the practical efficiency and value of the proposed controlled MADS unclear.

3. Marginal Performance Gains: Even optimized MADS configurations show only modest improvements over the single-agent baseline, especially on CommonsenseQA. These weak gains, combined with the lack of comparison to strong baselines like SC, make the overall performance results less convincing and highlight the need to demonstrate a clearer advantage to justify MADS complexity.

**Questions:**

no question

---

> ### Author Response · Authors · 2025-12-03
> **Reply to Reviewer cbtG**
>
> Dear Reviewer cbtG,
>
> Thanks for your insightful reviews and suggestions. We appreciate your recognition that the sycophancy persona dynamics is interesting to the community. We believe many of your concerns are addressable in the revised version.
>
> 1. **Limited Dataset Diversity**: We agree that evaluating additional task types (e.g., math, code) could further strengthen generalizability. Our intention, however, is not to provide general design principles across all task domains, but to establish an operational framework for analyzing and controlling sycophancy in multi-agent debate. We therefore chose two representative datasets—reasoning and factuality—where debate is known to yield gains and where sycophancy meaningfully interacts with disagreement dynamics. Expanding to more task formats would strengthen the findings in our work, but the core mechanisms we identify (e.g., sycophancy-driven disagreement collapse) are not tied to any particular dataset.
> 2. **Missing Comparison to Self-Consistency (SC)**: We view SC as complementary rather than directly comparable to multi-agent debate within our framework. SC improves performance through sampling diversity, whereas our focus is on understanding interaction dynamics between agents and isolating sycophancy as a tunable property within those dynamics. As such, our goal is not to demonstrate that MADS is universally superior to SC under fixed computational budgets, but to study how and why debate systems fail or succeed as sycophancy varies. Comparing SC is valuable when the objective is identifying the best performing algorithm, but our contribution is orthogonal: we provide tools for analyzing and controlling sycophancy in interacting agents, which SC does not involve.
> 3. **Marginal Performance Gains**: Our results highlight that performance improvements, even when modest, are systematically linked to sycophancy control. The objective of the paper is not to present a new state-of-the-art debate system, but to show that sycophancy plays a central role in failure modes such as disagreement collapse and that controlling it can reliably improve debate outcomes without model retraining. These findings are meaningful even when gains are subtle, because they reveal why debate can underperform and offer actionable insights for designing more stable multi-agent systems. Stronger performance gains may be achievable when integrating our framework with more advanced sycophancy-controlling strategies or training-based methods, which we consider an important direction for future work.
>
> Again, we’d like to express our gratitude for your insightful feedback!

---

### Official Review · Reviewer_q9a2 · 2025-11-04

**Soundness:** 2
**Presentation:** 4
**Contribution:** 1
**Rating:** 4
**Confidence:** 5

**Summary:**

This paper investigates sycophancy in multi-agent debating systems. The authors propose formal definitions and metrics for measuring “inter-agent sycophancy,” introduce a controlled persona spectrum, and evaluate its effects on debate outcomes across decentralized and centralized frameworks.

**Strengths:**

-   **[Problem importance]** The paper addresses an important aspect of multi-agent collaboration (sycophancy and agreement bias).

-   **[Novel perspective]** The framing is of sycophancy as not being innately "bad" is an interesting perspective that is not explored at all in the literature. This framing provides researchers a lens through which to better understand the role of sycophancy in multi-agent collaboration. While there are some issues with the paper (discussed later) this perspective, along with the formalism of the authors framework, is a solid selling point.

-  **[Formalism]**  The authors' framework for studying sycophancy is relatively formal and rigorous. This consistents of the introduction of well motivated metrics along with a "tunable" parameter quantifying sycophancy (although the actual tuning requires prompt-based heuristics).

-  **[Some nice analysis]** The paper presents clear empirical correlations (e.g., between sycophancy score and disagreement collapse).

**Weaknesses:**

Overall I really like the proposed framework and style of investigation, but the paper feels too incomplete to recommend acceptance. This is the type of paper that feels bad to reject; with more complete analysis and experiments I think this could be quite a strong contribution.

-  **[Limited empirical scope]:**: Only two models, two datasets, and two collaboration paradigms (both training-free). The primary contribution of this work is observational; a framework to study sycophancy and then observations made with in that framework (the tuning seems more like a consequence than a main contribution). As such, I would have expected the experiments and results to be far more comprehensive. Its very difficult for me to understand how general these trends are, or how useful the framework is, within this limited scope. Beyond just the "small" number of datasets, models, and baseline methods used, the baseline methods themselves are extremely simple and outdated. There have been a ton of followup works which patch many of the issues with the initial proposals of MAD.

- **[Lack of meaningful baselines]:** In the context of the above comment, I would have been very interested in seeing whether training-based methods for multi-agent debate leads to different observations. My intuition is that when the models are actually trained for this type of settings (whether its "argumentative" debate or "collaborative" debate) that sycophancy would be far less common and when it occurs it would be far more beneficial since the training would cause the models to learn a strong joint strategy. Given how different off-the-shelf models function in collaborative/argumentative settings compared to fine tuned models, this is a glaring omission from the current paper.

-   **[Depth of analysis is shallow]** Given that sycophancy is a relatively well observed phenomenon, the results seem to largely confirm expected trends rather than yielding new insights. For example, when we get to section 5.2, the authors it is hard to fully appreciate both the results associated with Figure 3 as well as the discussion around Debater Design Recommendations. Since the experiments don't look at the ways in which other works have dealt with sycophancy we have no idea whether the design recommendations are meaningful at all (maybe the effects of these recommendations are already subsumed by existing approaches, which are cited in the paper but not used). Similarly regarding Figure 3 and the assosiated discussion, its difficult to get into depth as to the actual effects of sycophancy when they aren't contextualized with other approaches.


-  **[Simple heuristic]** Beyond the issues with the scope and depth of observational experiments, the proposed mitigation (balancing) strategy is exceedingly simple. I like that it leverages the the earlier metric, but this heuristic is just another simple prompt engineering modification to debate. The strategy need to be complicated if it works well, but again its difficult to assess the efficacy with no meaningful baselines, so I can only judge the heuristic based on the amount of insight that it provides (which at face value is very little).

**Questions:**

See weaknesses

---

> ### Author Response · Authors · 2025-12-03
> **Reply to Reviewer q9a2**
>
> Dear Reviewer q9a2,
>
> Thanks for your valuable feedback on our paper. We’d like to thank you for your acknowledgment of the significance and novelty of our proposed framework for studying sycophancy in the multi-agent debating system. We believe many of your comments are addressable in the revised manuscript.
>
> 1. **Limited empirical scope**: We acknowledge that broader empirical coverage could further strengthen the study. Our objective, however, is not to provide general system design principles across all possible settings, but to introduce an operational framework for analyzing and designing debating systems through the lens of sycophancy—a factor that our results show to be strongly correlated with failures in handling disagreement within debate. Prior work has demonstrated that debate behaviors can vary substantially across model versions, datasets, and implementations [1]. Accordingly, our contribution focuses on isolating sycophancy as a key tunable parameter rather than attempting an exhaustive survey of every configuration.
> Our empirical choices were made to ensure representativeness while maintaining a clear analytic focus. We evaluate (1) two task types—reasoning and factuality—where debate is known to outperform single-agent generation, (2) two widely used open-source models from distinct model families (Qwen and Llama), and (3) two canonical debate paradigms that form the basis of many later variants. This design enables us to study sycophancy within the foundational communication structures of debate, without confounds introduced by highly specialized or rapidly evolving systems. Across all examined settings, we consistently find that sycophancy correlates with disagreement collapse and associated performance degradation, and that controlling sycophancy yields measurable improvements. We will release our evaluation and sycophancy-control code to support application of the framework to additional models, datasets, and debate frameworks.
>
> 2. **Lack of meaningful baselines**: We agree that comparisons with training-based mitigation strategies could provide additional context. However, such methods fundamentally differ in scope and assumptions from our prompting-based framework. Training-based systems reshape model behavior through supervised or RL-based tuning, whereas our goal is to provide a practical, model-agnostic framework for analyzing and controlling sycophancy without additional training requirements. Direct comparison would therefore be methodologically asymmetric and not directly informative about the mechanisms our framework aims to expose. Investigating how training interacts with sycophancy dynamics is therefore a natural and interesting direction for future work, but its omission does not invalidate our findings about sycophancy’s role in prompt-based multi-agent debate.
>
> 3. **Depth of analysis is shallow**: Our contribution lies in proposing an operational framework for analyzing and designing debate systems through the lens of sycophancy. While sycophancy is indeed a broadly observed phenomenon, there has been very limited work examining its mechanisms and operational implications specifically within multi-agent debate. To our knowledge, only one recent concurrent work directly studies sycophancy in multi-agent settings [2], and it approaches sycophancy primarily as a uniformly harmful behavior. In contrast, our results highlight that sycophancy has a more nuanced role: it can contribute to failure modes such as disagreement collapse, but it can also serve as a mechanism by which agents flexibly incorporate correct information from others. This distinction is central to our design recommendations and, to our knowledge, has not been previously articulated. The sycophancy mitigation methods proposed in that work are also training-based, whereas our focus is on prompt-based debate systems.

---

> ### Author Response · Authors · 2025-12-03
> **Continuation**
>
> 4. **Simple heuristic**: Our goal with the proposed balancing strategy is not to introduce a complex mitigation technique, but rather to demonstrate how our operational framework can directly inform actionable interventions. It shows that once sycophancy is explicitly measured and understood, even lightweight prompt-based adjustments can meaningfully influence debate dynamics. In this sense, the heuristic is an illustration of how the framework can be used, rather than a claim that it is the optimal or final approach. We agree that evaluating against additional baselines, especially training-based methods, could offer a broader picture of efficacy. However, our contribution is centered on establishing sycophancy as a key tunable factor and providing a practical, model-agnostic methodology for analyzing and adjusting it. Within this scope, the balancing strategy serves as a proof of concept demonstrating that sycophancy control can improve debate outcomes without retraining or architectural modifications.
> More sophisticated mitigation techniques, including training-based approaches, can certainly be explored within our framework. We view these extensions as valuable future work, but their absence does not diminish the conceptual contribution: showing that sycophancy can be operationalized, measured, and controlled in prompt-based multi-agent debate using simple and interpretable mechanisms.
>
> Thank you again for your constructive feedback!
>
> [1] Zhang, Hangfan, et al. "Stop Overvaluing Multi-Agent Debate--We Must Rethink Evaluation and Embrace Model Heterogeneity." arXiv preprint arXiv:2502.08788 (2025).
>
> [2] Pitre, Priya, Naren Ramakrishnan, and Xuan Wang. "Consensagent: Towards efficient and effective consensus in multi-agent llm interactions through sycophancy mitigation." Findings of the Association for Computational Linguistics: ACL 2025. 2025.

---

### Author Response · Authors · 2025-12-03
**Rebuttal Summary**

We thank all the reviewers for their constructive feedback. Across the comments, we identified one major theme—**experimental coverage((. Below, we summarize the key contributions of our paper and explain why its contributions remain solid.

Despite limited space for exhaustive empirical coverage, the paper presents **the first operational framework** that makes sycophancy in multi-agent debate systems measurable, controllable, and directly integratable into system design. Our goal is not to offer universal design principles for every debate setting, but to introduce a practical framework for analyzing and designing debating systems through the lens of sycophancy—a factor that our results show is strongly associated with failures in handling disagreement. Across all tested settings, sycophancy consistently correlates with disagreement collapse and performance degradation, and controlling it produces measurable gains. The framework is model-agnostic, scalable, and positioned as a foundation for future research on long-horizon and tool-using agents. We will also release our evaluation and sycophancy-control code to enable designers of multi-agent debating systems to determine the appropriate sycophancy configuration for their own setups.

We believe the revised version will directly address reviewer concerns while preserving the paper’s soundness and contributions.

---

### Note · Authors · 2026-01-06

I have read and agree with the venue's withdrawal policy on behalf of myself and my co-authors.